# Less Vibrotactile Feedback Is Effective to Improve Human Balance Control during Sensory Cues Alteration

**DOI:** 10.3390/s22176432

**Published:** 2022-08-26

**Authors:** Noémie Anctil, Zachary Malenfant, Jean-Philippe Cyr, Katia Turcot, Martin Simoneau

**Affiliations:** 1Faculté de Médecine, Département de Kinésiologie, Université Laval, Quebec, QC G1V 0A6, Canada; 2Centre Interdisciplinaire de Recherche en Réadaptation et Intégration Sociale (Cirris) du CIUSSS de la Capitale Nationale, Quebec, QC G1M 2S8, Canada; 3Faculté des Sciences et de Génie, Département de Génie Électrique et de Génie Informatique, Université Laval, Quebec, QC G1V 0A6, Canada

**Keywords:** upright standing, body sway, sensory augmentation, sensorimotor control, vibrotactile feedbacks

## Abstract

For individuals with altered sensory cues, vibrotactile feedback improves their balance control. However, should vibrotactile feedback be provided every time balance control is compromised, or only one-third of the time their balance is compromised? We hypothesized that vibrotactile feedback would improve balance control more when provided every time their balance is compromised. Healthy young adults were randomly assigned to two groups: group 33% feedback (6 males and 6 females) and group 100% feedback (6 males and 6 females). Vibrotactile feedbacks related to the body’s sway angle amplitude and direction were provided, while participants stood upright on a foam surface with their eyes closed. Then, we assessed if balance control improvement lasted when the vibrotactile feedback was removed (i.e., post-vibration condition). Finally, we verified whether or not vibrotactile feedback unrelated to the body’s sway angle and direction (sham condition) altered balance control. The results revealed no significant group difference in balance control improvement during vibrotactile feedback. Immediately following vibrotactile feedback, both groups reduced their balance control commands; body sway velocity and the ground reaction forces variability decreased. For both groups, unrelated vibrotactile feedback worsened balance control. These results confirmed that participants processed and implemented vibrotactile feedback to control their body sways. Less vibrotactile feedback was effective in improving balance control.

## 1. Introduction

The control of the human bipedal upright stance requires an accurate perception of body sway amplitude and direction. Body sway motion is estimated through sensory cues from the visual, vestibular, and proprioceptive systems [1,2]. According to the reliability of these cues, the brain performs a sensorimotor transformation, creating a corrective torque and reducing body sway amplitude with respect to the earth-vertical axis [3]. Thus, when sensory signals are altered or incongruent, body motion becomes less accurate. This leads to suboptimal sensorimotor transformation and impaired balance control [4,5,6,7,8]. Sensory augmentation provides relevant and additional cues about body sway amplitude and improves balance control [9,10]. Additional sensory cues can be provided through auditory, tactile, or visual systems. Multiple studies have reported improvement in balance control during sensory augmentation for healthy adults [11,12,13,14,15,16], healthy older adults [17], and in adults with vestibular dysfunction [13,18,19,20,21,22,23,24]. A reduction in the body sway amplitude, with additional sensory cues, is inversely related to the reliability of intrinsic sensory feedback. Therefore, populations with balance control impairment benefit the most from additional sensory cues [12,13,14,22]. Healthy young adults have little improvement potential, likely because of a ceiling effect [22]. Contrary to the visual and auditory stimuli, the tactile stimulus is preferred, as tactile cues applied on the trunk do not interfere with body sway-evoked sensory cues. Besides, tactile feedback does not interpose with the visual or auditory cues of everyday tasks [10]. Vibrotactile feedback is commonly applied to the trunk region [11,18,19,20,23,24], likely because trunk stability is a good predictor of balance [25,26] and the location and direction of vibrotactile stimulation applied to the trunk are accurately detected [27,28]. Vibrotactile feedback systems can improve the sensorimotor control of individuals with neurological diseases, the posture of musicians or arm guidance [29,30,31,32].

The processing of sensory augmentation cues implies necessary cognitive processes, as the brain must map body sway direction with the sensory feedback [33,34,35]. Although sensory augmentation improves balance control, researchers have reported that adding sensory cues can increase the cognitive load, decreasing the dual-task performance [17,36]. Compared to healthy young adults, the secondary cognitive task performance decrement is exacerbated in healthy older adults and individuals with vestibular dysfunction. Consequently, the quantity and nature of the feedback should be selected to provide enough sensory cues to improve balance control and to avoid cognitive overload. Still, it is unclear how often feedback should be provided to improve balance control. Thus, the novelty of the present study is to assess whether less frequent vibrotactile feedback can improve balance control. In addition, vibrotactile feedbacks were provided either during anteroposterior or mediolateral body sways, while visual and plantar sole mechanoreceptor cues were altered in healthy adults.

The main goal of this study was to determine if the quantity of vibrotactile feedback changed balance control improvement when visual and somatosensory cues were altered. We reasoned that vibrotactile feedback would strengthen the relationship between the balance motor commands and the estimation of body sway dynamics. Therefore, we hypothesized that balance control improvement would be scaled according to the quantity of feedback provided. The secondary goals were: (1) to determine if following vibrotactile feedback, balance control improvement carried over, (2) to examine if erroneous (sham) vibrotactile feedback altered balance control, and (3) to assess whether individuals with less effective balance control benefit the most from vibrotactile feedback. We reasoned that transient balance control improvement following vibrotactile feedback and increased body sways during erroneous vibrotactile feedback would indicate processing and implementing of the vibrotactile feedback. We further argued that individuals with poor balance control should benefit more from vibrotactile feedback.

## 2. Materials and Methods

### 2.1. Participants

Twenty-four healthy young adults, aged between 20 and 30 years old, volunteered for this study. The exclusion criteria included self-reported musculoskeletal, psychological, or neurological impairments or pregnancy. The participants’ level of physical activity (International Physical Activity Questionnaire (IPAQ) score, self-reported), anthropometric, and demographic characteristics did not differ between groups (Table 1, *p* values are for unpaired *t*-test). The study was conducted according to the guidelines of the Declaration of Helsinki and approved by the Institutional Review Board (CIUSSS de la Capitale Nationale—readaptation: Project #MP-13-2020-1874). Therefore, all participants gave their informed consent before they participated in the study.

### 2.2. Experimental Set-Up: Vibrotactile Feedback Device

Participants were asked to stand on the force platform (Model Optima HPS, Advanced Mechanical Technology Inc., Watertown, MA, USA) or on a 6 cm-thick foam mat (Airex AG, Sins, Argovie, Switzerland) placed over the force platform, with their eyes opened or closed. In some conditions, we chose to have the participants standing on the foam with their eyes closed to alter somatosensory and visual cues. Since altering sensory cues increases body sways amplitude, vibrotactile feedback should be helpful to reduce body sway amplitude and velocity.

We developed a vibrotactile feedback device providing tactile cues according to body sway angle and direction. The vibrotactile feedback device (Figure 1) implies an inertial measurement unit (IMU, ICM-20948, InvenSense, San José, CA, USA), a microcontroller (Arduino Mega 2560, Arduino LLC, Boston, MA, USA), and four vibration motors (Model No. 310-103, Precision Microdrives, London, UK). Participants wore a belt with four vibrator motors placed on the anterior, posterior, left and right trunk at the 12th thoracic (T12) vertebrae level. The vibration motors were directly in contact with the skin of the participants, securely fixed with tape and covered with an elastic band. The IMU was located slightly above the superior iliac crest (approximation of the center of mass, i.e., 4th lumbar (L4) vertebrae). Using the IMU, we calculated the body sway angle along both axes (i.e., AP and ML) using angular velocity and linear acceleration. For each trial, body sway kinematics were conveyed by the microcontroller to the hard disk and stored using MATLAB (version R2020a, MathWorks, Natick, MA, USA).

First, the angular velocity and linear acceleration data were low pass filtered (built-in digital filter) with a cut-off frequency of 5.7 Hz. Then, the body sway angles were calculated along the AP (Equation (1)) and ML (Equation (2)) axes:(1)$${\tilde{\theta }}_{\mathrm{AP}}={tan}^{-1}\left(\frac{a_{z}}{a_{x}}\right)$$
(2)$${\tilde{\theta }}_{\mathrm{ML}}={tan}^{-1}\left(\frac{a_{y}}{a_{x}}\right)$$
where ${\tilde{\theta }}_{\mathrm{AP}}$ (°) and ${\tilde{\theta }}_{\mathrm{ML}}$ (°) are an estimate of the body sway angles calculated between the vector representing the linear accelerations along the AP ($a_{z}$) or ML ($a_{y}$) axes and the vector representing the gravitational accelerations $a_{x}$. Then, a complementary discrete filter (i.e., a sensor fusion algorithm) used the raw acceleration, angular velocity, and body sway angle signals to improve the estimate of the body sway angles:(3)$$\theta_{\mathrm{AP}}=\alpha \times \left({\tilde{\theta }}_{\mathrm{AP}-1}+\omega_{z}\times \Delta t\right)+\left(1-\alpha \right)\times {\tilde{\theta }}_{\mathrm{AP}}$$
(4)$$\theta_{\mathrm{ML}}=\alpha \times \left({\tilde{\theta }}_{\mathrm{ML}-1}+\omega_{y}\times \Delta t\right)+\left(1-\alpha \right)\times {\tilde{\theta }}_{\mathrm{ML}}$$

The estimate of the body sway angles (i.e., ${\tilde{\theta }}_{\mathrm{AP}},{\tilde{\theta }}_{\mathrm{ML}}$) served as input in the complimentary filter (Equations (3) and (4)). In these equations, $\theta_{\mathrm{AP}}$ (°) (or $\theta_{\mathrm{ML}}$) is the calculated angle at the current iteration, ${\tilde{\theta }}_{\mathrm{AP}-1}$ (°) (or ${\tilde{\theta }}_{\mathrm{ML}-1}$) is the calculated angle at the previous iteration, $\omega_{z}$ (°/s) (or $\omega_{y}$) is the corresponding angular velocity, Δt (s) is the time step between each iteration (10 ms) and α is the weighting parameter between 0 and 1 (set to 0.78). The precision of the body sway angle was 0.02°. It represents the range of angle values when the IMU is set on a motionless surface.

Thresholds were defined along the AP and ML directions for each participant. The thresholds represented one standard deviation of the distribution of the body sway angle along the AP and ML direction, when the participants were standing with their eyes closed on a foam surface (Figure 2). When body sway angle was larger than the participant-specific threshold, one of the vibration motors was activated. Only one vibration motor could be activated at a time, and vibrotactile feedback was provided to the trunk in the direction of the body sway (Figure 3). For example, at ~10 s on panel A, the body sway angle along the AP direction exceeded the threshold. At this instant, vibrotactile feedback was delivered on the posterior part of the trunk (Figure 3—panel B). Approximately 1 s later, the body sway angle was greater than the threshold along the ML axis (Figure 3—panel C). Consequently, vibrotactile feedback was applied on the left part of the trunk (Figure 3—panel D). It is possible that the body sway angle was greater than the thresholds simultaneously along the AP and ML axes. In this circumstance, vibrotactile feedback was provided in the direction of the largest body sway angle. Vibrotactile feedback stopped when the body sway angle was smaller than the thresholds (dead zone). The frequency of the vibration motors was 250 Hz, and according to the manufacturer, the vibration amplitude was ~1.8 g for this frequency.

### 2.3. Protocol and Experimental Conditions

Before the experiment, participants were randomly assigned to either the 100% or the 33% feedback groups and completed the IPAQ—Short Form. This self-reported questionnaire assesses physical activity levels according to the intensity of physical activity performed in the last seven days as part of daily life. Participants were asked to stand still barefoot with their arms alongside. They wore noise-blocking headphones to avoid bias from vibration motors and environmental noise. The support base of the participants was standardized. The distance between both feet 5th metatarsal heads equaled the length of the feet. Overall, participants performed 70 trials, and each trial lasted 20 s. Right before each trial, while participants stood with their eyes open, a five-second-long calibration was completed to reset the IMU’s orientation according to the median value of each angle direction. To minimize fatigue, participants had seated rest periods every 10 trials. The experimental protocol contained two phases. First, there were four conditions (Figure 4—panel A): In the first condition (reference), participants performed five trials with their eyes open (EO) followed by five trials with their eyes closed (EC) while standing directly on the force platform (i.e., a hard surface (HS)). The abbreviations EOHS (i.e., eyes open on a hard surface) and ECHS (i.e., eyes closed on a hard surface) will be used hereafter. This condition served to quantify the baseline balance control performance of both groups. In the second condition (control), participants performed 10 trials while standing on a foam surface (FS) with their eyes closed. The abbreviation ECFS (i.e., eyes closed on a foam surface) will be used hereafter. The aim of this condition was to increase body sway by removing visual cues and reducing the reliability of the foot plantar sole mechanoreceptors. Further, we calculated the means and standard deviations of the body sway angles from the distribution of the body sway angle along the AP and ML axes of each participant. Based on these measures, we set the threshold for activating vibrotactile feedback to one standard deviation in the conditions following (Figure 2). In the third condition (vibrotactile feedback), participants performed 20 trials while standing on the foam surface with their eyes closed. If their body sway angle was larger than the threshold, vibrotactile feedback was delivered on their trunk according to the direction of body sway for at least 0.5 s. Participants in the 100% feedback group received vibrotactile feedback every time their body sway angle exceeded their threshold. In contrast, participants in the 33% feedback group received vibrotactile feedback for one in every three events that their body sway angle was greater than their threshold. In the fourth condition (post-vibrotactile), the participants performed 10 trials with eyes closed on a foam surface; however, there was no vibrotactile feedback. The aim of this condition was to assess if improvement in balance control lasted in the absence of vibrotactile feedback. Following pilot data, we elected to perform only 10 trials in the reference, control, and post-vibrotactile conditions to keep the duration of the experiment within 2 h. Despite many rest periods, we wanted to reduce physical and cognitive fatigue. Before the vibrotactile feedback condition, participants explored the functioning of the vibrotactile feedback device. While standing with their eyes opened on the foam surface, participants were asked to tilt their trunk in a backward, forward, right, and left direction. Thereafter, we asked them whether they noticed an association between the vibrotactile feedback and their body sway direction. All participants noticed the relation between the direction of the vibrotactile feedback and their body sway. They also observed that vibrotactile feedback was provided only if the amplitude of their body sways were large enough. Thereafter, the functioning of the vibrotactile feedback device was explained to participants. Thus, prior to performing the experiment, participants knew that moving in the opposite direction stopped the vibration. We elected to use repulsive cues, as balance improvement is better in this condition [37]. The second experimental phase (Figure 4—panel B) occurred ~5 min after the first phase. Participants were standing with their eyes closed on the foam surface. In this condition (sham), however, vibrotactile feedback was not related to the direction of the body sway. Vibrotactile feedback was randomly provided (i.e., any directions except the direction of large body sway) when the body sways were greater than the threshold. There were 20 trials in the sham condition. Participants were unaware that vibrotactile feedback was unrelated to body sway direction. The purpose of the sham condition was to verify if participants processed and used vibrotactile feedback to improve their balance control [16]. We reasoned that if participants processed vibrotactile feedback, their balance control would worsen. Otherwise, if incongruent vibrotactile feedback did not alter balance control, it would suggest that vibrotactile feedback was not processed. Such findings would be surprising as in order to reduce body sways, the brain needs an accurate estimate of the body sway dynamics [3,38]. An alternative hypothesis would be that participants detected that the vibrotactile feedback was unrelated to their body sway direction and ignored the feedback. To verify this hypothesis, after the sham condition, we asked the participants whether they noticed a difference in the vibration feedback, and if they changed their balance control strategy compared to the previous vibrotactile feedback condition.

### 2.4. Data and Statistical Analysis

To characterize the dynamics of the body sways, we analyzed the angle and angular velocity of the body sways along the AP and ML axes. To assess the neuromuscular control required to control body sways, we analyzed the forces applied on the ground along both axes. Body sways angle and angular velocity were calculated from the IMU data and the neuromuscular control from the ground reaction forces. IMU and force data were sampled at 100 Hz and synchronized. Data processing was performed using custom-made MATLAB scripts (The MathWorks R2020a, Natick, MA, USA). The IMU and force data were low pass filtered with a dual-pass 4th order Butterworth filter with a cut-off frequency of 6 Hz. We characterized balance control using measures that considered discrete events and average balance control. The average balance control was computed through an ellipse covering 85% of the area that was covered by the body sway angles and ground reaction forces, along the AP and ML axes. Average measures are insensitive to sudden change in the data. Thus, discrete events were considered by calculating the root mean square (RMS) values of the data along both axes.

Statistical analyses were conducted using Statistica (Version 7.1, Statsoft, Tulsa, OK, USA). The participants’ characteristics (age, weight, and height) in both groups were compared using an independent *t*-test for each characteristic. To assess the functioning of the vibrotactile feedback device, for each participant we calculated the means of the number and duration of the tactile vibration for vibrotactile feedback and sham conditions. Two separate independent T-test compared both groups. To assess if the vibrotactile thresholds differed across groups and along the AP and ML directions, a repeated measures analysis of variance (ANOVA) was performed (2 groups (100%, 33%) × 2 directions (AP, ML)). To verify whether balance control was alike between groups, in the reference and control conditions, we used an ANOVA with repeated measures (2 groups (100%, 33%) × 3 conditions (EOHS, ECHS, ECFS) × 2 directions). To confirm whether vibrotactile feedback enhanced balance control differently between groups, we ran a repeated measures ANOVA (2 groups (100%, 33%) × 2 conditions (ECFS, vibrotactile feedback) × 2 directions (AP, ML)) comparing the control and vibrotactile feedback conditions. We calculated Spearman’s correlation (MATLAB function *corr*() with type = Spearman) between the improvement in balance control (i.e., the percentage change between vibrotactile feedback and ECFS conditions) and balance control in the ECFS condition, to assess whether participants with poor balance control, when sensory cues were altered (i.e., ECFS condition), benefit the most from vibrotactile feedback. To determine if balance control improvement continued in the absence of vibrotactile feedback for both groups, we compared balance control during the last 10 trials of the vibrotactile feedback condition with the post-vibrotactile condition using a repeated measures ANOVA (2 groups (100%, 33%) × 2 conditions (vibrotactile feedback, post-vibrotactile) × 2 directions (AP, ML)). We calculated the means of the last 10 trials of the vibrotactile feedback condition, as there were 10 trials in the post-vibrotactile condition. Moreover, if vibrotactile feedback improved balance control, improvement had to be steadier in the last 10 trials. Finally, to assess whether each group used vibrotactile feedback, we compared balance control during the control and sham conditions using a repeated measures ANOVA (2 groups (100%, 33%) × 2 conditions (ECFS, sham) × 2 directions (AP, ML)). For all analyses, to decompose the significant interaction effects, we performed post-hoc analysis using Tukey’s honest significant difference (HSD) test. The significance level was set at *p* < 0.05. We elected to report the *p* value if the *p* value was ≥ 0.001, otherwise we reported *p* < 0.001. For significant results, we calculated eta squared, as it is a measure of effect size.

## 3. Results

### 3.1. Initial Conditions (Reference and Control Conditions)

The analysis of the RMS values of body sway angle and angular velocity in the reference and control conditions (Figure 5) revealed no group difference (main effect of the group: F(1,22) = 0.10, *p* = 0.82 and F(1,22) = 0.00, *p* = 0.94, respectively). The RMS values of the body sway angle and angular velocity were larger along the AP compared to the ML axes (main effect of direction: F(1,22) = 182.51, *p* < 0.001, η^2^ = 0.17 and F(1,22) = 118.90, *p* < 0.001, η^2^ = 0.11, respectively). Altering visual and plantar sole mechanoreceptor cues did not increase body sway differently between groups (main effect of condition: F(2,44) = 313.72, *p* < 0.001 and F(2,44) = 519.8, *p* < 0.001, for body sway angle and angular velocity, respectively). We observed a significant interaction between direction and condition for body sway angle and angular velocity (interaction direction by condition: F(2,44) = 39.23, *p* < 0.001, η^2^ = 0.04 and F(2,44) = 20.85, *p* < 0.001, η^2^ = 0.01 respectively). The decomposition of the interactions revealed that the RMS values of the body sway angle and angular velocity were larger along the AP axis than the ML axis for all conditions (ps < 0.01). Along the AP axis, the RMS values of the body sway angle and angular velocity were larger in the absence of vision on the hard surface (ps < 0.05, for ECHS compared to the EOHS conditions) and larger when standing on the foam surface in the absence of vision (ps < 0.001, for ECFS versus EOHS and ECHS). For the RMS values along the ML axis, the body sway angle and angular velocity were not significantly different in the reference conditions (*p* = 0.98 and *p* = 0.76, for EOHS versus ECHS, respectively) and were larger when standing on the foam surface in the absence of vision (ps < 0.001, for ECFS versus EOHS and ECHS).

The analysis of the RMS values of the ground reaction forces for the reference and control conditions (Figure 6) revealed no group difference (main effect of group: F(1,22) = 2.10, *p* = 0.16). For both groups, the RMS values of the ground reaction forces were larger along the AP compared to the ML axes (main effect of direction: F(1,22) = 52.40, *p* < 0.001, η^2^ = 0.03). The RMS values were different across conditions (main effect of condition: F(1,22) = 425.10, *p* < 0.001, η^2^ = 0.86). A significant interaction between direction and condition was observed (interaction direction by condition: F(2,44) = 20.9, *p* < 0.001, η^2^ = 0.005). Decomposition of the interaction revealed that the RMS values of the ground reaction forces were larger along the AP compared to the ML axes for all conditions (ps < 0.001). For both groups, when standing on the hard surface, removing vision increased the RMS value along the AP axis (*p* < 0.001, EOHS versus ECHS) but not along the ML axis (*p* = 0.05, EOHS versus ECHS). Along both axes, the RMS values were larger when both groups stood on the foam surface with the eyes closed, compared to when they stood on the hard surface (ps < 0.001, for ECFS compared to EOHS and ECHS).

### 3.2. Vibrotactile Thresholds

Vibrotactile thresholds (Table 2) showed no difference between groups (main effect of group: F(1,22) = 0.01, *p* = 0.92). For both groups, the threshold values were larger along the AP compared to the ML axes (main effect of direction: F(1,22) = 7.06, *p* = 0.01). No significant interaction was observed between the groups and direction (interaction group by direction: F(1,22) = 0.60, *p* = 0.45).

### 3.3. Vibrotactile Feedback Condition

Balance control in the vibrotactile feedback compared to the control conditions (Figure 7) revealed no difference between the groups for the RMS values of the body sway angle and angular velocity (main effect of group: F(1,22) = 0.16, *p* = 0.70 and F(1,22) = 0.17, *p* = 0.68, respectively). The RMS values of the body sway angle and angular velocity were larger along the AP compared to the ML axes (main effect of direction: F(1,22) = 6.75, *p* = 0.02, η^2^ = 0.13 and F(1,22) = 18.04, *p* < 0.001, η^2^ = 0.23, respectively). The RMS values for the vibrotactile feedback condition were smaller compared to the control condition (i.e., ECFS) for the body sway angle (main effect of condition: F(1,22) = 14.52, *p* < 0.001, η^2^ = 0.06). The RMS values for the angular velocity did not differ between conditions (main effect of condition: F(1,22) = 4.08, *p* = 0.056). For either the body sway angle or the angular velocity, no significant interaction was observed (ps > 0.05).

Spearman’s correlation between the improvement in balance control with vibrotactile feedback and balance control in the control condition (ECFS) revealed weak correlations for the RMS values of the body sway angle (33% feedback group: AP axe, r = −0.07, *p* = 0.83 and ML axe, r = −0.15, *p* = 0.65; 100% feedback group: AP axis, r = −0.59, *p* < 0.05 and ML axe, r = −0.29, *p* = 0.35). Overall, a significant correlation was only observed in the AP direction for the 100% feedback group.

The comparison of the RMS values of the ground reaction forces between groups for the control condition and the vibrotactile feedback condition (Figure 8) showed no group difference (main effect of group: F(1,22) = 1.11, *p* = 0.30). Larger RMS values were observed along the AP compared to the ML axes (main effect of direction: F(1,22) = 31.83, *p* < 0.001, η^2^ = 0.18). The analysis of the RMS values revealed larger ground reaction forces variability in the vibrotactile feedback compared to the control conditions (main effect of condition: F(1,22) = 28.06, *p* < 0.001, η^2^ = 0.14). No significant interaction was observed between these factors (*p* > 0.05).

The unpaired *t*-tests for the number and duration of tactile vibrations (Table 3) revealed differences between the groups. As expected, the 100% feedback group received more tactile vibration than the 33% feedback group, and the sum of the vibration duration was longer, confirming the good functioning of the vibrotactile device.

### 3.4. Post Vibrotactile Condition

The comparison of balance control between the vibrotactile feedback and the post-vibrotactile conditions (Figure 9) revealed no group difference for the RMS values of the body sway angle and angular velocity (main effect of group: F(1,22) = 0.38, *p* = 0.54 and F(1,22) = 0.94, *p* = 0.34, respectively). The RMS values were larger along the AP compared to the ML (main effect of direction: F(1,22) = 10.62, *p* = 0.004, η^2^ = 0.14 and F(1,22) = 35.66, *p* < 0.001, η^2^ = 0.27, for the body sway angle and angular velocity, respectively). Compared to the vibrotactile condition, no change in the body sway angle and a decrease in the body sway angular velocity was observed in the post-vibration condition (main effect of condition: F(1,22) = 3.02, *p* = 0.10 and F(1,22) = 22.05, *p* < 0.001, η^2^ = 0.04, respectively). A significant direction by the condition interaction was observed only for the body sway angle (interaction direction by condition: F(1,22) = 6.19, *p* = 0.02, η^2^ = 0.02). The decomposition of the interaction revealed that the body sway angle was larger for the post-vibration condition along the AP axis (ps < 0.02) and no difference was revealed in the ML direction between conditions.

The analysis of the variability in the ground reaction forces between the vibrotactile and post-vibrotactile conditions (Figure 10) revealed no group difference for the RMS values of the ground reaction forces (main effect of groups: F(1,22) = 2.22, *p* = 0.15). However, for both groups, a larger variability was observed along the AP compared to the ML axes (main effect of direction: F(1,22) = 38.10, *p* < 0.001, η^2^ = 0.20). The analysis revealed smaller variability in the ground reaction forces for the post-vibration compared to the vibrotactile feedback conditions (main effect of condition: F(1,22) = 55.39, *p* < 0.001, η^2^ = 0.23). We observed no interaction between these factors (ps > 0.05).

### 3.5. Sham Condition

It is worth mentioning that following the sham condition, we questioned the participants to verify whether they noticed that the vibrotactile feedback was unrelated to body sway direction. Remarkably, only one participant felt that the vibrotactile feedback did not match to the direction of the body sway. All participants, however, reported having more difficulty in controlling their body sways compared to the vibrotactile feedback condition.

The comparison of balance control between the control condition (ECFS) and the sham condition (Figure 11), revealed no group difference (main effect of group: F(1,22) = 0.09, *p* = 0.77 and F(1,22) = 0.12, *p* = 0.74, for the body sway angle and angular velocity, respectively). The RMS values of the body sway angle and angular velocity were larger along the AP compared to the ML axes (main effect of direction: F(1,22) = 6.16, *p* = 0.02, η^2^ = 0.04 and F(1,22) = 15.59, *p* < 0.001, η^2^ = 0.09, respectively). In the sham condition, the RMS values of the body sway angle and angular velocity were larger than for the control condition (main effect of condition: F(1,22) = 4.31, *p* < 0.05, η^2^ = 0.09 and F(1,22) = 5.50, *p* = 0.03, η^2^ = 0.08, respectively). We observed no significant interaction between these factors (ps > 0.05).

The analysis of the variability of the ground reaction forces between the control condition (ECFS) and the sham condition (Figure 12) revealed larger RMS values in the sham condition (main effect of condition: F(1,22) = 14.05, *p* = 0.001, η^2^ = 0.17). For both groups, larger variability was observed along the AP compared to the ML axes (main effect of direction: F(1,22) = 33.73, *p* < 0.001, η^2^ = 0.09). There was no significant group difference (main effect of group: F(1,22) = 1.27, *p* = 0.27) and interaction between these factors (*p* > 0.05).

Results from the unpaired t-tests for the number and duration of the tactile vibration (Table 4) confirmed that the 100% feedback group received more vibration. The sum of the duration of these tactile vibrations was longer than the 33% feedback group for the sham condition.

### 3.6. Ellipse Data

In addition to reporting the variability in the body sways and ground reaction forces, we calculated the average balance performance across the experimental condition by fitting an ellipse covering 85% of the body sway angle, angular velocity or ground reaction forces along the AP and ML axes. The results of the average balance response were similar to the RMS values. These results are reported in the Appendix A.

## 4. Discussion

Results from the current study revealed that balance control improvement was similar between the groups receiving frequent (i.e., 100% feedback group) and less frequent (i.e., 33% feedback group) vibrotactile feedback. For both groups, following vibrotactile feedback (post-vibration condition), the improvements in balance control lasted. When vibrotactile feedback was unrelated to the direction of the body sways (sham condition), the balance control of both groups worsened. The maintenance of balance control improvement during the post-vibrotactile feedback condition, and the worsened balance control during the sham condition, confirmed that the participants processed and implemented vibrotactile feedback. Contrary to our hypothesis, individuals with poor balance control did not benefit more from vibrotactile feedback when sensory cues were altered. Overall, these results confirmed that less frequent vibrotactile feedback can improve balance control in healthy young adults. Less frequent vibrotactile feedback can be more beneficial, as it likely implies less cognitive load. Nonetheless, this suggestion should be verified in individuals with altered balance control [36].

### 4.1. Control Condition

To conduct this study, twenty-four healthy young adults were randomly assigned to either the 100% feedback or the 33% feedback groups. We compared the balance control of both groups to ensure that any potential improvement in the balance control, during the vibrotactile feedback condition, did not reflect group differences in the baseline balance control. As expected for healthy young adults, we observed no group difference either in the absence of visual cues or when plantar sole mechanoreceptors and visual cues (condition ECFS) were simultaneously altered. Individuals with severe sensory loss benefit more from sensory augmentation than others with lesser sensory deficits [39]. Consequently, we expected that participants with poorer balance control would improve their balance more with vibrotactile feedback in the ECFS condition. This hypothesis was not confirmed. When altering the sensory cues, the range of body sway angles was small, reducing the potential for balance control improvement with vibrotactile feedback. For healthy individuals, proper functioning of the sensorimotor loops enables detection and correction of larger body sways.

### 4.2. Quantity of Vibrotactile Feedback

Sensorimotor adaptation is an iterative correction of the motor commands based on movement errors [40]. Therefore, we reasoned that signaling large body sways more frequently should lead to a better mapping between the estimation of body sway kinematics and balance motor commands, leading to improved balance control. Contrary to our hypothesis, the decrease in the variability of the body sway angle was similar for both groups during the vibrotactile feedback condition. An alternative hypothesis could be that too frequent extrinsic feedback alters the processing of intrinsic body sway-evoked sensory cues, causing poor balance control [41]. None of these hypotheses was supported by our results. To reduce the variability in body sway angles, the group with less frequent feedback (i.e., the 33% feedback group) could occasionally rely on intrinsic body sway-evoked sensory cues, while the 100% feedback group likely processed and implemented the vibrotactile feedback. The current results, however, suggest that both groups combined vibrotactile feedback and body sway-evoked sensory cues to improve balance control. If the 33% feedback group had relied solely on intrinsic feedback, their balance control would not have worsened in the sham condition. Thus, the participants likely processed the vibrotactile feedback. When no vibrotactile feedback was provided despite large body sways, a reduction in body sway variability suggests that participants likely processed body sway-evoked sensory cues to improve balance control. For the 100% feedback group, frequent vibrotactile feedback did not block the processing of body sway-evoked sensory cues. Otherwise, balance control would have worsened in the post-vibration condition. Consequently, participants in the 100% feedback group likely mixed vibrotactile feedback and body-sway evoked sensory cues to improve balance control. The absence of a group difference in balance control improvement might suggest that the device provided the same amount of vibrotactile feedback to both groups. The 100% feedback group, compared to the 33% feedback group, received more vibrotactile feedback. The duration of these feedbacks was longer, confirming the effective functioning of the vibrotactile feedback device. The current results suggest that both groups could switch between intrinsic and extrinsic feedback to improve balance control despite sensory cues alteration. For individuals with functioning sensorimotor loops, body sway-evoked sensory cues can compensate for the reduced quantity of vibrotactile feedback.

For both groups, improvement in balance control caused greater changes in the ground reaction forces. This observation confirmed that both groups altered their balance motor commands. Large or frequent changes in the balance motor commands, however, may cause muscular fatigue. The increase in the variability in the ground reaction forces was similar for both groups, supporting the absence of group difference in the vibrotactile feedback condition. Although participants in the 33% feedback group received less vibrotactile feedback, they likely processed body sway-evoked sensory cues and implemented proper changes in their balance motor commands to reduce body sways. Overall, the increase in neuromuscular control during vibrotactile feedback should be considered when using vibrotactile feedback devices to improve the balance control of individuals with balance instability. Long-term use of vibrotactile feedback, however, might enhance sensorimotor control by promoting the weighting of less variable sensory cues and reduce balance motor commands’ variability [10,42].

### 4.3. Post-Vibrotactile Feedback

Although providing feedback commonly improves performance, some studies reported performance decrement when feedback was removed. When feedback is too frequent, it can cause a dependency [41,43], thus removing feedback sometimes impairs sensorimotor performance. Improvement in balance control, observed during the vibrotactile feedback, was maintained when feedback was turned off (post-vibration condition) only along the ML direction. Extrinsic feedback involves cognitive processes and sensorimotor reweighting mechanisms [10]. The brain can process vibrotactile feedback as an additional sensory cue, and assign a large weight to this additional cue or to body sway-evoked sensory cues. The lasting balance control improvement along the ML direction indicates that either volitional or non-volitional cognitive processes alerted the sensorimotor mechanisms about the larger than usual body sways along the frontal plane. When standing on a hard surface with normal stand width, the body sways along the AP direction are larger than along the ML direction [44]. In contrast to normal standing, standing on the foam surface without visual cues caused a larger than usual body sway along the ML direction, evoking important sensory cues. This increase probes the sensorimotor mechanisms to closely control body sways along the frontal plane. Ballardini et al. [16] reported lasting balance control improvement along the AP direction for participants standing on a stable hard surface in the absence of visual cues. In their study, however, vibrotactile feedback was provided only in the AP direction. The vibrotactile feedback may be beneficial only when the sensorimotor loops are functioning since, in individuals with vestibular dysfunction, balance control improvement disappeared rapidly when the vibrotactile feedback was removed [24]. Proper functioning of the sensorimotor loops likely enhances the interplay between intrinsic and extrinsic feedbacks.

Contrary to the vibrotactile feedback condition, the decrease in the ground reaction forces variability suggests that participants generated less corrective torque as the body sway evoked sensory cues. Such a reduction in the changes of balance motor commands was reported in individuals with bilateral vestibular loss standing on a compliant surface in the absence of visual cues. Providing auditory and vibrotactile feedback led these individuals to decrease muscle activity [45]. Consequently, long-term training with vibrotactile feedback may enhance the mapping between the perception of body sway kinematics and the sensorimotor mechanisms, leading to an efficient association between body sways and balance motor commands.

### 4.4. Sham Vibrotactile Feedback

In accordance with the study of Ballardini et al. [16], when vibrotactile feedback was unrelated to body sway directions (sham condition), we observed an increase in the variability of the body sway angles, velocity, and ground reaction forces. This observation suggests that balance control improvement was caused by processing and implementing feedback rather than merely increasing alertness. When asked whether they observed a change in the relationship between the vibrotactile feedback and their body sways, only one participant indicated that feedback was unrelated to body sway directions. Therefore, vibrotactile feedback may have enhanced the sensory reweighting mechanism by selecting sensory cues with less variance rather than using a volitional cognitive strategy. A non-volitional cognitive strategy can direct attention to body sway-evoked sensory cues, strengthening the mapping between body sway detection and the balance motor commands.

### 4.5. Limitations

Only one quiet standing balance task was evaluated in our study. Results may be different during a dynamic task such as walking. A greater proportion of falls occurs when walking [46,47]. Therefore, future research should verify whether less frequent vibrotactile feedback could improve balance control during walking. Instead of providing feedback related to body sway angles, feedback would be delivered according to trunk angular velocity along the frontal plane, as this is an indicator of dynamic balance during walking [25,26,48].

The current device delivered vibrotactile feedback on four torso locations. More vibration motors covering additional body sway directions and different vibration intensities. According to the amplitude or velocity of body sway, this may lead to better balance control improvement [19,23]. Nonetheless, the vibrotactile device and current experimental protocol were designed to minimize cognitive load. Previous studies reported a decreased performance in a secondary cognitive task (i.e., a longer reaction time) while vibrotactile feedback was provided [17,36,49]. Furthermore, the decrement in the cognitive task was more important for healthy older adults and adults with vestibular dysfunction than healthy adults. Therefore, it is crucial to provide feedback that improves balance control without loading the cognitive processes.

In the current study, participants performed the sham condition after the vibrotactile feedback condition. Thus, participants expected vibrotactile feedback related to body sway direction and amplitude. During the sham condition, participants used necessary vibrotactile feedback, as suggested by the decreased performance. Consequently, performing the vibrotactile feedback condition prior to the sham condition may have worsened balance control in the sham condition. We elected to add a sham condition to assess whether vibrotactile feedback was processed and implemented to improve balance control. Without the sham condition, the question could not be answered.

During the vibrotactile condition, the amplitude of neuromuscular control increased, potentially inducing muscle fatigue. For healthy young adults, however, we observed a reduction in the variability of the ground reaction forces when the feedback was turned off. Future studies need to evaluate if long-term use of less frequent vibrotactile feedback improves balance control in populations with balance control impairment, without causing large changes in the balance motor commands. In the current study, healthy young adults stood on a foam surface with their eyes closed. This unusual condition likely required an increase in neuromuscular control.

## 5. Conclusions

This study compared two quantities of vibrotactile feedback on the balance control of young healthy adults, when the reliability of somatosensory and visual cues was reduced. The results suggest that less frequent vibrotactile feedback effectively improves balance control. Although we did not assess the cognitive performance, less vibrotactile feedback should likely reduce cognitive load. The worsening of balance control when vibrotactile feedback was unrelated to body sway direction suggests that participants processed and implemented the vibrotactile feedback. Future studies should assess whether vibrotactile feedback, provided at specific times during dynamic readaptation exercises, could improve balance control. This could translate into reducing the risk of falling and enhancing the quality of life of individuals with balance control disorders.

## Figures and Tables

**Figure 1 sensors-22-06432-f001:**
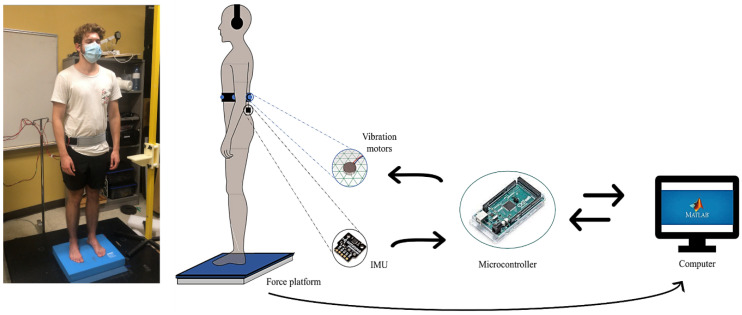
Experimental set-up. The participant stood still on a force platform (or on a foam placed on the force platform) wearing headphones and the vibrotactile feedback device, which implied an IMU located at the level of the superior iliac crest. A microcontroller processed IMU data and activated or deactivated the vibration motors according to body sway amplitude and direction. IMU and force platform data were acquired using MATLAB. The picture on the left depicts a participant during the pilot study. The vibration motors are under the t-shirt directly in contact with the skin of the participant. They are securely fixed with tape and covered with an elastic band, while the gray elastic belt holds the IMU. Note, this pilot participant did not wear headphones, but all other participants wore headphones.

**Figure 2 sensors-22-06432-f002:**
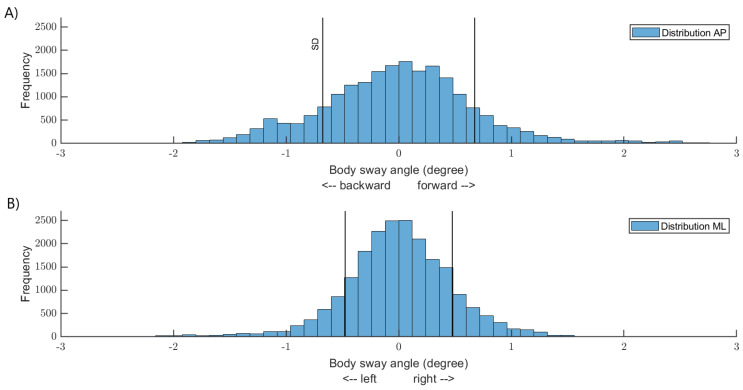
Distribution of the body sway angle of a representative participant standing on the foam surface with eyes closed. Panel (**A**) body sway angle along the AP axis. Panel (**B**) body sway angle along the ML axis. On both panels, vertical black lines depict one standard deviation (SD), that is, the participant-specific threshold to provide vibrotactile feedback.

**Figure 3 sensors-22-06432-f003:**
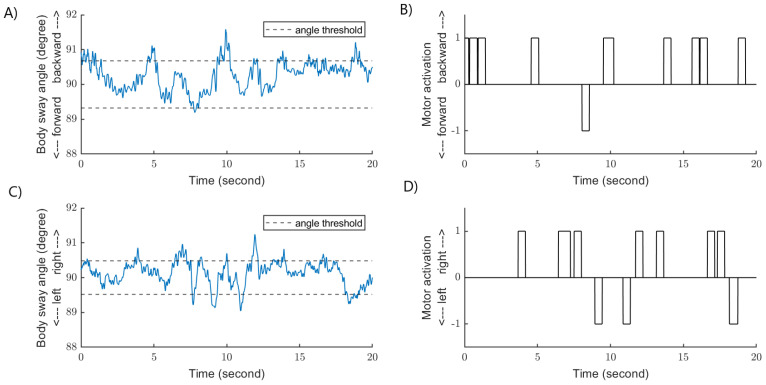
Examples of real-time body sway angle (blue lines, panel (**A**,**C**)) and the thresholds (black dashed lines) for a representative participant. Panel (**A**,**B**)) shows body sway along the AP axis and corresponding activation (1) and deactivation (0) of the vibration motors. Panel (**C**,**D**)) shows body sway along the ML axis and corresponding activation (1) and deactivation (0) of the vibration motors. The vibration motors were activated in the direction of body sway when the body sway angle was larger than the participant-specific threshold. Vibrotactile stimulation lasted for a minimum of 0.5 s.

**Figure 4 sensors-22-06432-f004:**
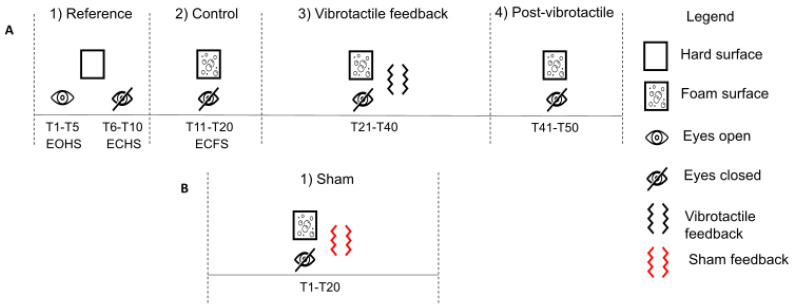
Experimental protocol. There were five conditions divided into two phases (**A**,**B**). Trials were either performed with eyes opened or closed while standing on a hard or foam surface. In addition, the participants had either no vibrotactile feedback (reference, control, and post-vibrotactile), vibrotactile feedback related (vibrotactile feedback) or unrelated (sham) to body sway direction. The order of the trials in each condition is indicated for each condition (i.e., T refers to trial, T first trial—T last trial).

**Figure 5 sensors-22-06432-f005:**
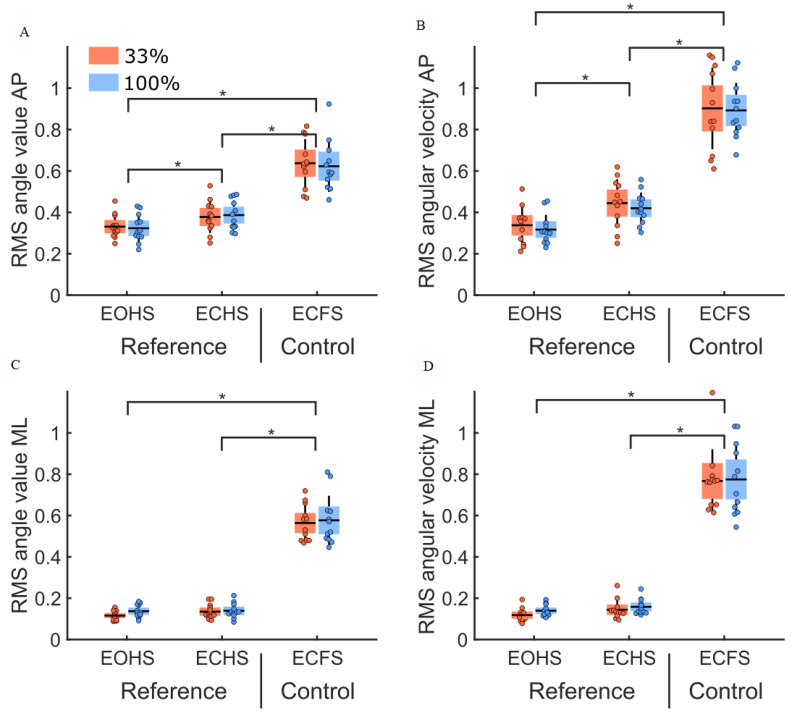
RMS values for body sway angle and angular velocity for both groups (33% and 100% feedback groups, red and blue boxes, respectively) for the reference condition (EOHS and ECHS) and for the control condition (ECFS). Panel (**A**,**C**) depict RMS values of the body sway angle along the AP and ML axes. Panel (**B**,**D**) depict RMS values of the body sway angular velocity along the AP and ML axes. The dots represent the mean results for each participant, horizontal lines depict the group’s means, boxes represent the group’s standard error of the mean, and vertical lines, one standard deviation. The units for the RMS values of the body sway angle and angular velocity are (°) and (°/s). The asterisks (*) indicate significant differences between conditions.

**Figure 6 sensors-22-06432-f006:**
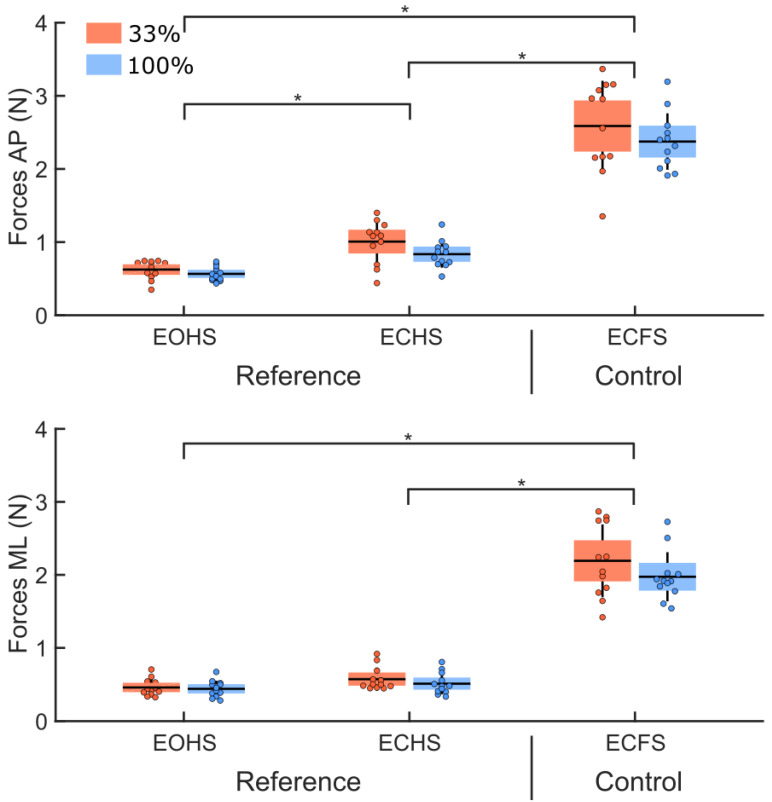
RMS values of the ground reaction forces along the AP (upper panel) and ML (lower panel) axes for both groups (33% and 100% feedback groups, red and blue boxes, respectively). Means are for the reference condition (EOHS and ECHS) and for the control condition (ECFS). The dots represent mean results for each participant, horizontal lines depict the group’s means, boxes represent the group’s standard error of the mean, and vertical lines, one standard deviation. The asterisks (*) indicate significant differences between conditions.

**Figure 7 sensors-22-06432-f007:**
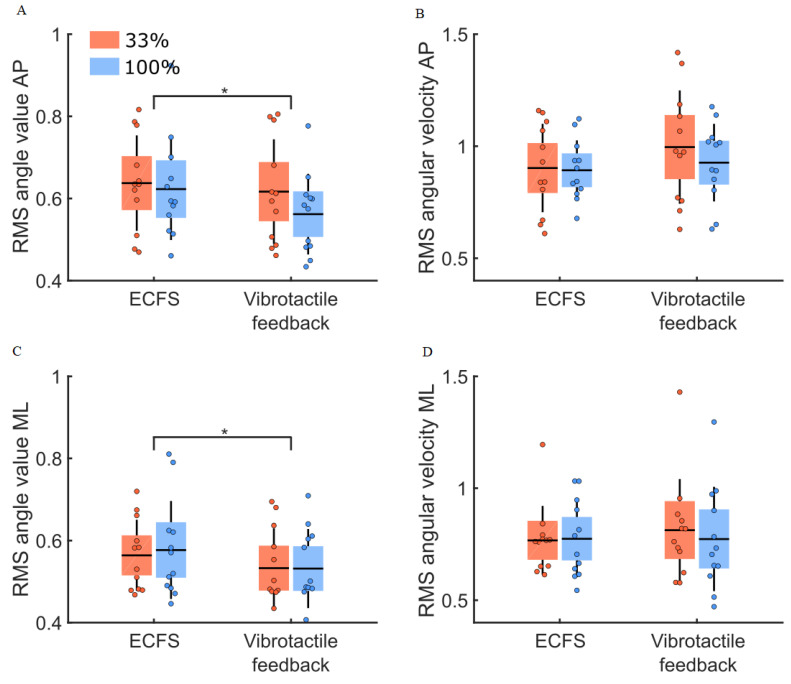
RMS values for body sway angle and angular velocity for both groups (33% and 100% feedback groups, red and blue boxes, respectively) in the control (ECFS) and the vibrotactile feedback conditions. Panels (**A**,**C**), RMS values of the body sway angle along the AP and ML axes. Panel (**B**,**D**), RMS values of the body sway angular velocity along the AP and ML axes. The dots represent the mean results for each participant, horizontal lines depict the group’s means, boxes represent the group’s standard error of the mean, and vertical lines, one standard deviation. The asterisks (*) indicate a significant main effect of condition. The units for the RMS values of the body sway angle and angular velocity are (°) and (°/s).

**Figure 8 sensors-22-06432-f008:**
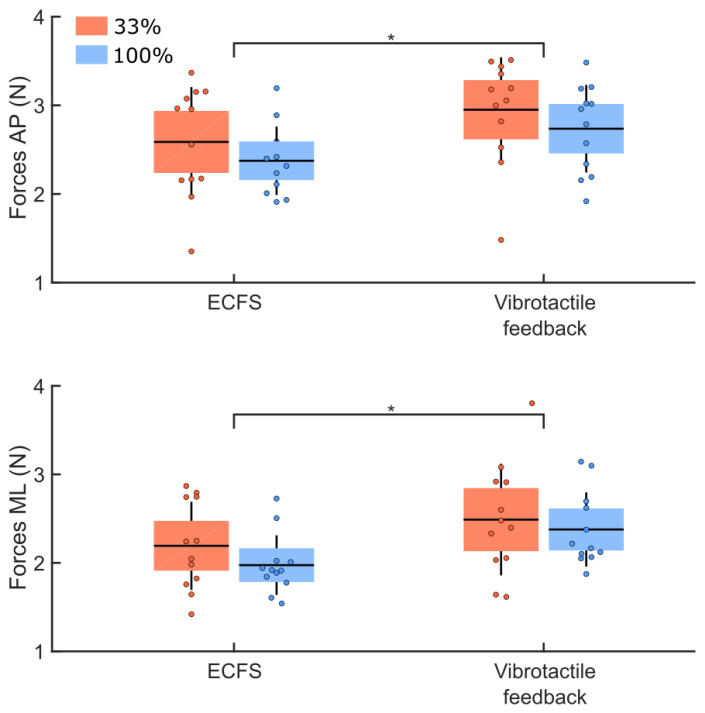
RMS values of the ground reaction forces along the AP (upper panel) and ML (lower panel) axes for both groups (33% and 100% feedback groups, red and blue boxes, respectively). These RMS values are for the control (ECFS) and the vibrotactile feedback conditions. The dots represent the mean results for each participant, horizontal lines depict the group’s means, boxes represent the group’s standard error of the mean, and vertical lines, one standard deviation. The asterisks (*) indicate significant main effect of condition.

**Figure 9 sensors-22-06432-f009:**
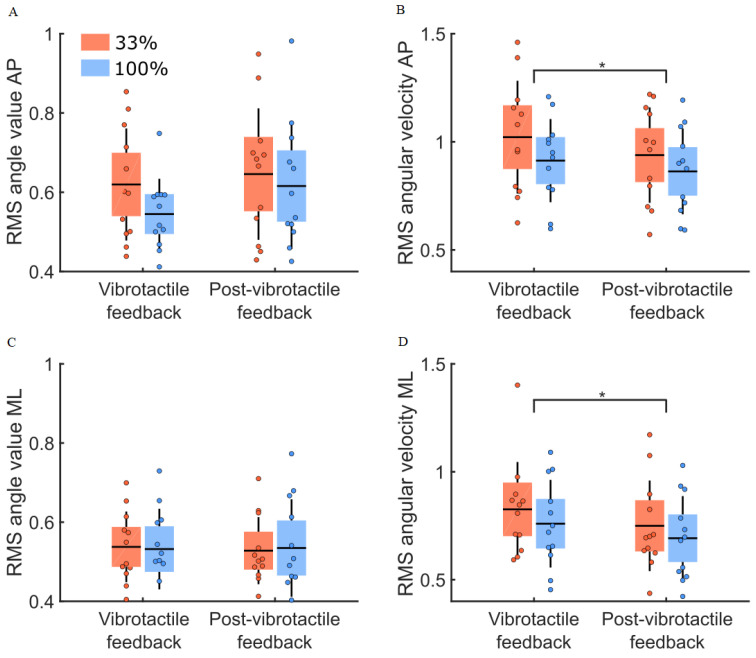
RMS values for body sway angle and angular velocity for both groups (33% and 100% feedback groups, red and blue boxes, respectively) for the vibrotactile (last 10 trials mean) and the post-vibrotactile conditions. Panels (**A**,**C**) depict RMS values of the body sway angle along the AP and ML axes. Panel (**B**,**D**) present the RMS values of the body sway angular velocity along the AP and ML axes. The dots represent the mean results for each participant, horizontal lines depict the group’s means, boxes represent the group’s standard error of the mean, and vertical lines, one standard deviation. The asterisks (*) indicate a significant main effect of condition. The units for the RMS values of the body sway angle and angular velocity are (°) and (°/s).

**Figure 10 sensors-22-06432-f010:**
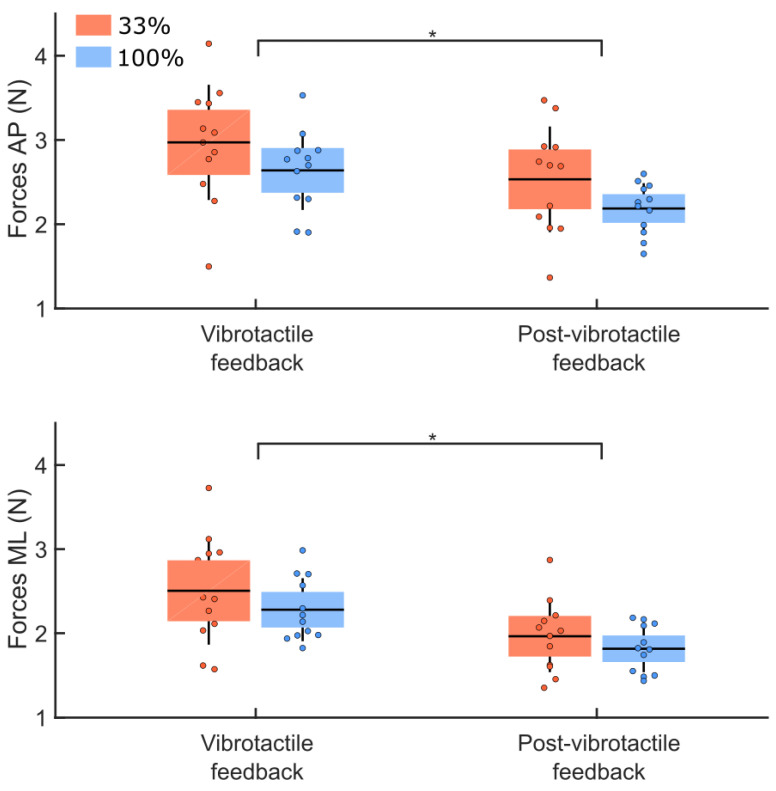
RMS values of the ground reaction forces along the AP (upper panel) and ML (lower panel) axes for both groups (33% and 100% feedback groups, red and blue boxes, respectively). These data are for the vibrotactile (last 10 trials mean) and the post-vibrotactile conditions. The dots represent the mean results for each participant, horizontal lines depict the group’s means, boxes represent the group’s standard error of the mean, and vertical lines, one standard deviation. The asterisks (*) indicate a significant main effect of condition.

**Figure 11 sensors-22-06432-f011:**
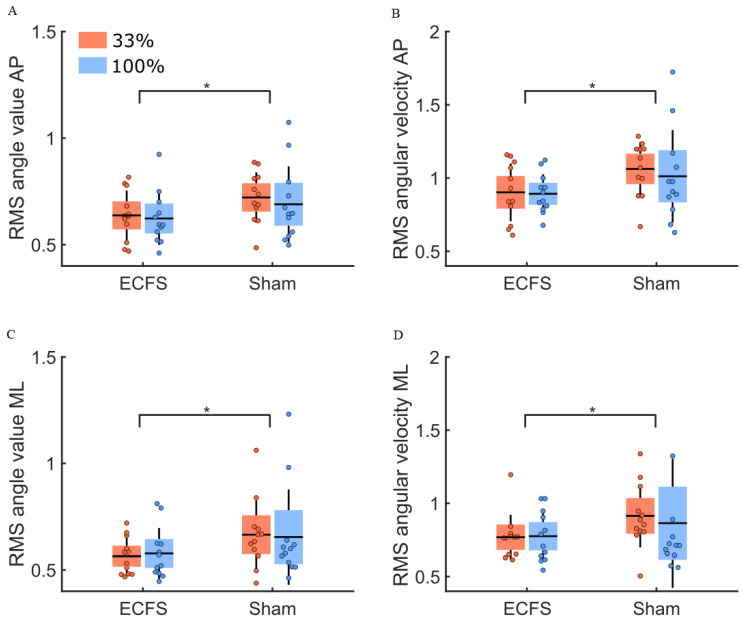
RMS values for body sway angle and angular velocity for both groups (33% and 100% feedback groups, red and blue boxes, respectively) for the control condition (ECFS) and the sham condition. Panels (**A**,**C**) depict the RMS values of the body sway angle along the AP and ML axes. Panel (**B**,**D**) show the RMS values of the body sway angular velocity along the AP and ML axes. The dots represent the mean results for each participant, horizontal lines depict the group’s means, boxes represent the group’s standard error of the mean, and vertical lines, one standard deviation. The asterisks (*) indicate a significant main effect of condition. The units for the RMS values of the body sway angle and angular velocity are (°) and (°/s).

**Figure 12 sensors-22-06432-f012:**
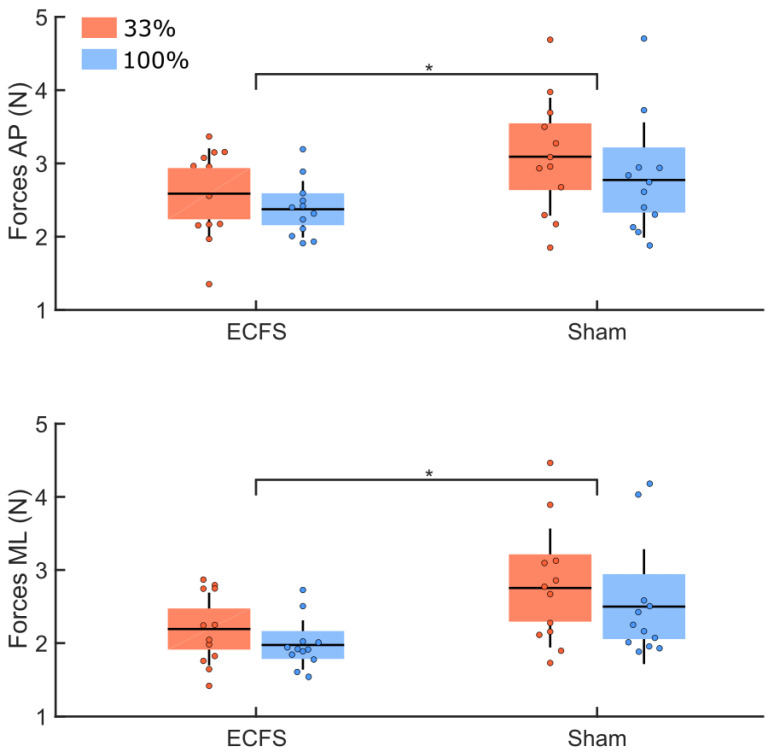
RMS values of the ground reaction forces along the AP (upper panel) and ML (lower panel) axes for both groups (33% and 100%, red and blue respectively). These data are for the control condition (ECFS) and the sham condition. The dots represent the mean results for each participant, horizontal lines depict the group’s means, boxes represent the group’s standard error of the mean, and vertical lines, one standard deviation. The asterisks (*) indicate a significant main effect of condition.

**Table 1 sensors-22-06432-t001:** Mean ± standard deviation of the morphological and demographic characteristics and reported physical activity of the participants of each group.

Group	100% Feedback	33% Feedback	*p*
Men:Women	6:6	6:6	
Age (yrs)	23.8 ± 2.5	24.6 ± 2.2	0.69
Weight (kg)	65.3 ± 12.0	74.3 ± 14.2	0.60
Height (m)	168.2 ± 7.7	171.7 ± 10.4	0.32
Score IPAQ (High:Moderate:Low)	9:1:2	9:2:1	

**Table 2 sensors-22-06432-t002:** Group means (± standard deviation) for the vibration threshold along the AP and ML axes.

	100% Feedback	33% Feedback
AP axis (°)	0.65 ± 0.13	0.67 ± 0.13
ML axis (°)	0.60 ± 0.12	0.58 ± 0.10

**Table 3 sensors-22-06432-t003:** Group means (±standard deviation) and statistics (i.e., t and *p* values) for the number and duration of tactile vibrations for the vibrotactile feedback condition.

Group	100% Feedback	33% Feedback	t-Value	*p*
Number of tactile vibrations	22.74 ± 3.86	9.10 ± 0.93	t(22) = 11.91	<0.001
Duration of tactile vibration (s)	13.68 ± 1.65	5.80 ± 0.71	t(22) = 15.24	<0.001

**Table 4 sensors-22-06432-t004:** Group means (± standard deviation) and statistics (i.e., t and *p* values) for the number and duration of tactile vibrations for the sham condition.

Group	100% Feedback	33% Feedback	t-Value	*p*
Number of vibrations	18.97 ± 2.97	9.21 ± 0.79	t(22) = 10.97	<0.001
Duration of vibration (s)	15.38 ± 2.21	7.67 ± 1.12	t(22) = 10.78	<0.001

## Data Availability

The data that support the findings of this study are available on reasonable request from the corresponding author. The data are not publicly available due to privacy or ethical restrictions.

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
