# Peer review of "Less Vibrotactile Feedback Is Effective to Improve Human Balance Control during Sensory Cues Alteration"

_sensors, 2022, doi:10.3390/s22176432_

Round 1

Reviewer 1 Report

The paper presents a study on the effects of vibrotactile feedback to improve balance control during sensory cues alteration. The paper is interesting and easy to read. However, the following points need to be improved before considering the paper suitable for publication.

1) The main novelties of the paper should be better highlighted with respect to the present literature. Furthermore, the main research question of the study should be better highlighted in the text.

2) It is not clear why the percentage of 33% was choosen. Has any preliminary tests been conducted to define this value?

3) More details on the participants involved in the test should be improved in the manuscript. How the participants were instructed before the test?

4) A picture of the experimental setup with a participant performing the test should be added to the manuscript.

5) It is not clear if the vibrations were used in attractive (move towards the vibration) or in repulsive (move away from the vibration) modes.

6) It is unclear whether a training effect occurred in the tests.

7) The amplitude and frequency of the vibrations of the vibrating actuators should be added in the manuscript.

8) The literature review should be improved by considering more recently published works on the topic. Some suggested articles are reported in the following:

Scalera, L., Seriani, S., Gallina, P., Di Luca, M., Gasparetto, A. (2018). An experimental setup to test dual-joystick directional responses to vibrotactile stimuli. IEEE Transactions on Haptics11(3), 378-387.

Yunus, R., Ali, S., Ayaz, Y., Khan, M., Kanwal, S., Akhlaque, U., Nawaz, R. (2020). Development and testing of a wearable vibrotactile haptic feedback system for proprioceptive rehabilitation. IEEE Access8, 35172-35184.

García-Valle, G., Arranz-Paraíso, S., Serrano-Pedraza, I., Ferre, M. (2020). Estimation of torso vibrotactile thresholds using eccentric rotating mass motors. IEEE Transactions on Haptics14(3), 538-550.

Reviewer 2 Report

This study aimed to assess the effects of different frequency of vibrotactile feedback on static postural control in young adults. I see some value in this study, but as it currently stands, it does not meet the standard for publication. Please consider the following:  

1.      It appears that there is text missing in 3.2 and 3.5 sections. Indeed, you talk about worsened balance in sham condition, but I see no results pertaining to that. After adding the additional text, please double check that all figure numbers, in-text reference to figures, and figure labels are correct.

2.      Consider adding some details on participant recruitment – were they students of your faculty, how did they get to know about the study?

3.      Line 148. Is 1SD threshold usual in this type of studies? Please comment briefly here and add references if possible

4.      Protocol section includes a figure that very nicely summarizes the experimental protocol – nevertheless, I miss some additional text to clarify some issues – how long were the trials, how many in each condition, the breaks in between?

5.      What were the instruction to the participants - a) in general, were they told to be as stable as possible, and b) were they told about the purpose of vibrotactile device?

6.      I miss effect size statistics, which would be more informative than the F values. Moreover, when you say that a change was similar in both groups, you could provide mean % change.

Minor comments:

-        Line 15: include n of males and females, and age range for the participants

-        Line 23: “altered” does not indicate the direction of the effect – improved or worsened?

-        Line 37 and 45: the citations should be merged

Round 2

Reviewer 1 Report

The paper was improved with respect to the previous version. However, a picture of a participant performing the test and a picture of the experimental setup are still missing. Furthermore, I also suggest adding the supplementary material to the paper as an appendix at the end of the manuscript.

Author Response

We thank the Reviewers for this second quick review of our paper.

The paper was improved with respect to the previous version. However, a picture of a participant performing the test and a picture of the experimental setup are still missing. Furthermore, I also suggest adding the supplementary material to the paper as an appendix at the end of the manuscript.

Response: As mentioned in our previous response to this comment, we do not judge that adding a picture of a participant will help the readers in understanding the experimental set-up. Nonetheless, as requested by this Reviewer, we added a picture of a person being tested during piloting. This explains why the participant on this picture did not wear headphone; we were piloting to check the quality of the signals. The person in the picture is a member of our research team. He accepted that we took a picture. The Reviewer must recognize that to take picture of a participant, researchers must justify to the Institutional Review Board (IRB) why a picture of a participant is required. The IRB will question the need of such picture. When presenting this project to the IRB, we did not plan to take a picture of a participant. Finally, the results previously in the supplementary material section are now in Appendix A.

Reviewer 2 Report

All comments have been addressed. 

Author Response

We thank the Reviewers for this second quick review of our paper.